# Context Dependent Sulf1/Sulf2 Functional Divergence in Endothelial Cell Activity

**DOI:** 10.3390/ijms23073769

**Published:** 2022-03-29

**Authors:** Tiago Justo, Nicola Smart, Gurtej K. Dhoot

**Affiliations:** 1Department of Comparative Biomedical Sciences, Royal Veterinary College, University of London, London NW1 OTU, UK; tjusto@rvc.ac.uk; 2Department of Physiology, Anatomy & Genetics, University of Oxford, South Parks Road, Oxford OX1 3PT, UK; nicola.smart@dpag.ox.ac.uk

**Keywords:** Sulf1, Sulf2, angiogenesis, VEGF cell signalling, TGFβ cell signalling

## Abstract

Signalling activities are tightly regulated to control cellular responses. Heparan sulfate proteoglycans (HSPGs) at the cell membrane and extracellular matrix regulate ligand availability and interaction with a range of key receptors. SULF1 and SULF2 enzymes modify HSPG sulfation by removing 6-O sulfates to regulate cell signalling but are considered functionally identical. Our in vitro mRNA and protein analyses of two diverse human endothelial cell lines, however, highlight their markedly distinct regulatory roles of maintaining specific HSPG sulfation patterns through feedback regulation of HS 6-O transferase (HS6ST) activities and highly divergent roles in vascular endothelial growth factor (VEGF) and Transforming growth factor β (TGFβ) cell signalling activities. Unlike Sulf2, Sulf1 over-expression in dermal microvascular HMec1 cells promotes TGFβ and VEGF cell signalling by simultaneously upregulating HS6ST1 activity. In contrast, Sulf1 over-expression in venous ea926 cells has the opposite effect as it attenuates both TGFβ and VEGF signalling while Sulf2 over-expression maintains the control phenotype. Exposure of these cells to VEGF-A, TGFβ1, and their inhibitors further highlights their endothelial cell type-specific responses and integral growth factor interactions to regulate cell signalling and selective feedback regulation of HSPG sulfation that additionally exploits alternative Sulf2 RNA-splicing to regulate net VEGF-A and TGFβ cell signalling activities.

## 1. Introduction

Quantitatively and qualitatively regulated cell signalling plays a key role in normal cardiovascular development and disease [1,2]. Coordinated cell signalling activation not only requires rapidly changing levels of multiple interacting growth factors but also the involvement of multiple co-receptors to regulate different stages of angiogenesis, cell proliferation, migration, and differentiation to assemble and expand blood vessels of an appropriate calibre. The sulfation pattern of heparan sulfate proteoglycans (HSPGs), one of the key co-receptors required for most cardiovascular cell signalling activities, can further modulate pathway activities to regulate blood vessel growth. For example, 6-O heparan sulfates of HSPGs, initially regulated by HS 6-O sulfotransferases [3], positively facilitate VEGF and FGF cell signalling during angiogenesis, whilst such cell signalling is inhibited by 6-O desulfation of HSPGs by two extracellular sulfatases described as Sulf1 and Sulf2 [4,5,6,7,8,9]. HS 6-O desulfation, on the other hand, may facilitate other cell signalling pathways such as Wnt signalling [8,10,11], while the effects of 6-O desulfation on pathways such as BMP or TGFβ have been reported but with conflicting findings [12,13,14,15,16]. Furthermore, Sulf1 and Sulf2 themselves may have varied effects in different cell types or under certain conditions, both in VEGF signalling, as the most prominent angiogenic pathway, as well as other cell signalling pathways proposed to control angiogenesis [2]. VEGF is critical for the development, growth, and survival of blood vessels [17], and HS 6-O sulfates have been reported to regulate angiogenic responses of endothelial cells to VEGF as well as FGF signalling since 6-O de-sulfation reduces endothelial cell sprouting and tube formation. Consequently, there has been considerable interest in the function and deployment of SULF1 and SULF2 activities due to their potential to modulate cell signalling to inhibit or promote growth [4,7,18,19]. This is further supported by their high level of expression during early development and disease, offering the possibility of targeting Sulfs therapeutically in angiogenesis and stem cell stimulation during vascular repair or regeneration. Despite sharing an identical molecular function, Sulf1 and Sulf2 demonstrate either overlapping or unique patterns of expression in specific tissues to promote or inhibit specific cell signalling pathways. However, their precise individual roles in different contexts are still unclear, although there has been some speculation about their possible differing roles in some neuronal and tumour tissues [20,21,22,23,24].

Our earlier study demonstrated high levels of Sulf1 and Sulf2 expression in developing cardiomyocytes and the endothelial lining of coronary blood vessels, which was downregulated during later development, to persist at a low level in endothelial cells but not in cardiomyocytes at adult stages [18]. The levels of both Sulf1 and Sulf2 are regionally upregulated in the injured heart following myocardial infarction in different cells [25] or the same cells [18]. Dynamic changes in Sulf1 and Sulf2 expression, leading to changes in VEGF and TGFβ cell signalling, also become apparent during in vitro hypoxia of HMec1 microvascular endothelial cells [18]. The present study further demonstrates that TGFβ and VEGF-A cell signalling is up-regulated following Sulf1 over-expression in HMec1 cells, mimicking the earlier response to hypoxia, before further changes in Sulf1/Sulf2 dynamics during later stages. Since both TGFβ and VEGF cell signalling play critical roles in cardiovascular biology, a better understanding of Sulf1 and Sulf2 functions and their regulatory roles in different cell types and contexts may reveal more appropriate therapeutic targets to improve outcomes following cardiovascular disease. Sulfs could act as rapid transient therapeutic targets, but we need to establish if and how these enzymes regulate one another or substitute for each other under relevant disease conditions. The present study demonstrates marked differences in the activities of Sulf1 and Sulf2 in endothelial cell activity and reveals complex interaction, in that they not only facilitate or inhibit different signalling pathways but also regulate overall sulfation patterns by regulating upstream sulfotransferase activities. Apart from enzymatic regulation of HSPG sulfation to control diverse cell signalling activities, chemical heparan sulfate is routinely used for both in vitro and transient in vivo clinical application following blood clotting detection in which the contribution of different endothelial cell types is not so well characterised.

## 2. Results

### 2.1. Sulf Over-Expression in HMec1 Cells Defines Distinct Sulf1 and Sulf2 Regulatory Roles

The effects of Sulf1 and Sulf2 over-expression in HMec1 human dermal microvascular endothelial cells were investigated using Sulf1 or Sulf2 plasmid transfection, compared with EGFP and/or pcDNA3 transfection, described here as controls. Cells for immunocytochemical analyses thus included transfections with pcDNA3 or EGFP (control) and full-length human Sulf1 (HS1) or full-length Sulf2 (HS2) DNA only, whereas RT PCR analyses also included two additional transfections with shorter inactive splice variants of Sulf1 (HS1-678,18) and Sulf2 (HS2-678,18) that lacked exons 6, 7, 8 and 18, as previously described [21,22]. The stable over-expression of Sulf1 and Sulf2 in HMec1 cells was confirmed using RT PCR (not shown) as demonstrated for other cell lines in our earlier studies [21,22] and immunocytochemical analyses (Figure 1). The levels of both Sulf1 and Sulf2 in untreated or EGFP or pcDNA3 transfected control HMec1 cells were generally quite low, but marked up-regulation of these proteins was observed following Sulf1 and Sulf2 transfections (Figure 1). Sulf enzymes hydrolyse 6-O sulfates of HSPGs for extracellular editing that are initially added by HS 6-O-sulfotransferases but the levels of HS6ST1, HS6ST2 or HS6ST3 were barely detectable in control HMec1 cells. Interestingly, Sulf1, but not Sulf2, over-expression in HMec1 cells resulted in marked up-regulation of HS6ST1, which was clearly apparent by both RT PCR and immunocytochemical analyses (Figure 1). In contrast, HS6ST2 or HS6ST3 did not significantly change following Sulf transfections.

### 2.2. Over-Expression of Sulf1, but Not Sulf2, in HMec1 Cells Augments TGFβ and VEGF-A Cell Signalling

We first examined if Sulf1 or Sulf2 over-expression in HMec1 cells leads to modification of TGFβ or BMP cell signalling using RT PCR and immunocytochemical analyses. RT PCR analysis clearly demonstrated the up-regulation of Smad2 and Smad3 as well as the TGFβ type I receptor (TGFβRI) activin receptor-like kinase 5 (ALK5) in Sulf1 over-expressing cells, while no such change was apparent in Sulf2 over-expressing cells (Figure 2). TGFβ1 was detected in all conditions, with the highest levels in Sulf1-overexpressing and control cells and slightly reduced levels in Sulf2 over-expressing cells. TGFβRII expression was minimally affected by Sulf1 over-expression but was markedly reduced by over-expression of Sulf2 as well as the shorter variants of Sulf1 and Sulf2 (Figure 2). The immunocytochemical staining for phospho-Smad2/Smad 3 in Sulf1 over-expressing HMec1 cells not only showed increased expression in cytoplasm but also at variable levels in the majority of nuclei. Unlike Smad2/3, no Smad1 or Smad5 expression was apparent in control or transfected HMec1 cells using RT PCR (not shown), although low-level expression of these proteins could be detected in a very small number of cells but did not change in response to either Sulf1 or Sulf2 over-expression (Figure 2). The differential effects on Smad2/3 versus Smad1/5 suggest that Sulf1 selectively enhances TGFβ but not BMP signalling in HMec1 cells, while Sulf2 minimally affects signalling through these pathways.

The impact of Sulf overexpression was also evaluated on VEGF-A signalling, one of the key angiogenic growth factors [1]. Although Sulf1 and Sulf2 have both been reported to inhibit angiogenesis, it was surprising to see marked up-regulation of VEGF-A in Sulf1 transfected cells and a modest increase in Sulf2 over-expressing cells by RT PCR (Figure 3). Based on the PCR product size, this slight increase in Sulf2 over-expressing cells related to VEGF_120_ (lacking the heparin-binding domain) only, while Sulf1 over-expression led to increased expression of both VEGF_120_ and VEGF_165_ isoforms. We also investigated the role of Sulf1 over-expression in HMec1 cells on Angiopoietin-1 (Ang1) and Ang2 ligands for the receptor tyrosine kinase Tie2 also considered essential for vascular development [1]. Ang1 expression was barely detectable in any samples using RT PCR in HMec1 cells, unlike Angiopoietin 2 (Ang2), which was detectable in all samples with the exception of those transfected with the shorter inactive Sulf variants (Figure 3) but with increased expression in Sulf1 transfected samples. Neuropilin 1 (NRP1), a multifunctional receptor of many endogenous cytokines including VEGF-A, also showed marked up-regulation in only Sulf1 transfected cells when compared with control or Sulf2 transfected cells. The expression of Tie1 and Tie2 receptors was apparent in all samples, with slightly variable levels in different samples (Figure 3). The surprising activation of VEGF signalling in Sulf1 transfected cells using RT PCR was further confirmed by its activation at the protein level using immunocytochemical analyses showing marked VEGF-A up-regulation (Figure 3), similar to RT PCR analysis. Ang1 up-regulation in Sulf1 transfected cells was much more easily apparent with antibody staining than RT PCR analysis, unlike Ang2 showing a marked increase using both these analytical procedures (Figure 4). Immunocytochemistry also showed marked increases in both NRP1 and NRP2 in Sulf1 transfected HMec1 cells, although their low levels of expression were also apparent at baseline in control and Sulf2 transfected cells (Figure 3).

### 2.3. Sulf1 and Sulf2 Over-Expression Leads to Diminished Proliferation and Cell Migration in HMec1 Cells

We next sought to determine the functional impact of manipulating Sulfatase levels, as a downstream effect of signalling alterations using phase contrast microscopy. Sulf1/Sulf2 transfections in HMec1 cells showed changes in cell morphology as the Sulf1 transfected cells often appeared aggregated or clustered, unlike control or Sulf2 transfected cells, although the latter were generally smaller in size (Figure 4A1). Since Sulf1 over-expression in HMec1 cells demonstrated selective VEGF and TGFβ activation, compared with Sulf2, we also investigated their effect on cell proliferation using the sulforhodamine B (SRB) method described by Vichai & Kirtikara [26] to measure cell density. Since Sulf over-expression markedly affected the morphology of HMec1 cells (Figure 4A1), all counting procedures will have limitations as the increased or decreased cell size may not equate with increased or decreased cell number. Nevertheless, SRB staining and colorimetric readings of transfected cells indicated a significant reduction in cell growth in both Sulf1 and Sulf2 transfected cells, with the reduction being much greater for Sulf2 compared with Sulf1 over-expression (Figure 4A2).

To examine if Sulf over-expression influences HMec1 cell migration, we used a wound-healing assay. When assessing wound closure after 8 h, this assay clearly demonstrated a significant reduction in cell migration in both Sulf1 and Sulf2 transfected cells when compared with the control cells (Figure 4B1, B2).

### 2.4. HMec1 Cell Responses to Increased VEGF, TGFβ or Their Inhibitors Highlights Their Interdependent Cell Signalling and Sulf1, Sulf2 Regulation

Since Sulf1 over-expression in HMec1 cells markedly upregulated VEGF and TGFβ expression, we next examined the effect of exposing control non-transfected HMec1 cells to VEGF-A or TGFβ1, or their inhibitors, for 48 h on Sulf1/Sulf2 signalling pathways that they affect. Included in these treatments were an exposure to 8 mM sodium sulfate and 8 mM sodium chlorate to compare the effects of global chemical sulfation and desulfation. None of the treatments induced any change in Sulf1 expression pattern, with the exception of exposure to the VEGFR2 inhibitor SU1498, which significantly induced Sulf1 expression revealed using RT PCR (Figure 5). In contrast, a marked induction of Sulf2 was observed following all treatments, with the exception of SU1498 treatment that did not upregulate but instead maintained one of the two shorter Sulf2 splice variants observed in untreated control HMec1 cells (Figure 5). Moreover, a striking diversity in Sulf2 splice variants was observed, as detected by RT PCR of the catalytic domain (verified by sequencing of amplified fragments). With the exception of untreated control and SU1498 treated HMec1 cells, all treated cells express full-length active Sulf2 to varying levels. The highest levels of shortest inactive Sulf2 splice variant 2 [27] were observed in control and SU1498 treated cells, while the highest level of medium-length Sulf2 splice variant was apparent in HMec1 cells exposed to TGFβ but no expression in SU1498 treated cells (Figure 5). The exposure of HMec1 cells to VEGF_165_ clearly showed the highest VEGF expression as expected, but with no detection of VEGF_120_ but VEGF was also detected in response to TGFβ exposure as well as under exposure to SU6668 and SU1498 inhibitors. While the highest level of Smad2 was observed following VEGF exposure, some Smad2 increase was also observed following not only TGFβ exposure but also global sulfation with 8 mM sulfate but not desulfation using 8 mM chlorate.

### 2.5. Sulf1/Sulf2 Over-Expression in ea926 Cells and Their Effect on HS 6-O Sulfotransferases 1,2,3

We also used another cell line, ea926, for Sulf1 and Sulf2 transfections to determine if all endothelial cells respond similarly or distinctly. Transfection of ea926 cells with Sulf1 and Sulf2 plasmids led to their increased expression in these cells, as expected, although, unlike HMec1 cells, ea926 cells already expressed significant levels of SULF1 and SULF2 proteins (Figure 6). RT PCR analysis showed high levels of HS6ST1 expression in all cells, but slightly higher levels were observed in Sulf1 and Sulf2 transfected cells that did not appear to reflect such distinction at the protein level using an immunocytochemical procedure. RT PCR analysis of HS6ST2 mRNA showed the highest level in control cells but reduced levels in Sulf1 and particularly Sulf2 transfected cells. HS6ST3 expression was apparent in only control and Sulf1 transfected cells (Figure 6).

### 2.6. Sulf1 Over-Expression, but Not Sulf2, Reduces TGFβ and VEGF Cell Signalling in ea926 Cells

We then examined if Sulf1 or Sulf2 over-expression in ea926 cells similarly modified TGFβ or VEGF cell signalling, as was apparent for HMec1 cells. RT PCR analysis showed high levels of Smad2 mRNAs in all samples but an indication of slightly higher levels in control and Sulf2 transfected cells when compared with Sulf1, and shorter Sulf variant transfections demonstrating lower levels of these transcripts. A similar but generally lower level of Smad3 expression was observed following Sulf1 and Sulf2 transfections of ea926 cells when compared with Smad2. ALK5 did not show such Sulf1/Sulf2 differences as at least low-level expression was apparent in all samples, with higher levels in control and Sulf1 transfected cells. Unlike HMec1 cells, ea926 cells also showed significant BMP signalling in ea926 cells as Smad1 and Smad5 expression were clearly apparent in control and Sulf2 transfected ea926 cells, as was the expression of ALK1 (Figure 7). The levels of Smad1, Smad5 and ALK1, in contrast, were much lower in Sulf1 transfected cells. TGFβ1 and TGFβRII expression with variable levels were also apparent in all samples. Higher Smad1, Smad5 expression in control and Sulf2 transfected ea926 cells, compared with Sulf1 transfected cells, were also confirmed at the protein level using immunocytochemical staining (Figure 7).

The effect of Sulf1 and Sulf2 over-expression on VEGF cell signalling was also examined in ea926 cells using two sets of primers, one of which detects both VEGF_120_ and VEGF_165_. VEGF expression observed in control ea926 cells was clearly maintained in Sulf2 transfected ea926 cells but markedly inhibited in Sulf1 transfected cells, with much reduced or undetectable VEGF expression by RT PCR (Figure 8). VEGF reduction in Sulf1 transfected cells compared with control and Sulf2 transfected cells was also indicated, albeit less clearly, by immunocytochemical analysis (Figure 8). NRP1 was undetectable in Sulf1 transfected cells when compared with high levels in Sulf2 transfected cells and the control cells, while Ang1, Ang2, NRP2, Tie1 and Tie2 were unchanged upon Sulf transfection (Figure 8). In keeping with this, Ang1 and Ang2 immunocytochemistry did not indicate obvious differences in this cell line (Figure 8).

### 2.7. The Effect of Sulf1 or Sulf2 Over-Expression in ea926 Cells on Growth and Cell Migration

Cell density measurements of control versus Sulf1 and Sulf2 transfected ea926 cells showed some reduction in cell growth following both Sulf1 and Sulf2 transfections, although the level of reduction was higher in Sulf1 over-expressing cells than Sulf2 over-expressing cells using the SRB staining procedure (Figure 9).

The effect of Sulf1 and Sulf2 overexpression on ea926 cell migration was also examined using a wound-healing assay. This protocol demonstrated reduced migration rates in both Sulf1 and Sulf2 transfected cells, although the reduction in Sulf1 was greater than in Sulf2 transfected ea926 cells (Figure 9).

### 2.8. Exogenous VEGF and TGFβ Exposure also Regulates Sulf1 and Sulf2 Expression in ea926 Cells

Since Sulf1 over-expression in ea926 cells appeared to inhibit or reduce VEGF and TGFβ cell signalling, we examined if the exposure of control untreated ea926 cells to VEGF and TGFβ or their inhibitors for 48 h led to changes in Sulf1/Sulf2 or the activities of these cell signalling pathways as well as their responses to global chemical sulfation and desulfation. While Sulf2, Smad2 and Smad3 levels were increased in all treatments with the exception of TGFβ, Sulf1 levels following such treatments showed markedly variable quantitative responses. For example, the highest level of Sulf1 expression was observed in response to VEGF_165_ exposure and global sulfation with some reduction in response to TGFβ and receptor tyrosine kinase inhibitor treatment (Figure 10). In contrast to HMec1 cells, expression of the shorter variants of Sulf 2 did not vary to a great extent; all Sulf2-expressing samples contained predominantly full-length isoform and low-level expression of the shorter variants.

## 3. Discussion

Both Sulf1 and Sulf2 are believed to inhibit VEGF activity, similar to their inhibition of FGF cell signalling [6,19,28], but Sulf1 over-expression by transfection in HMec1 cells led to increased VEGF expression, just as hypoxia induces Sulf1 increase followed by VEGF up-regulation [18]. It is possible that VEGF expression induced by hypoxia or Sulf1 transfection fundamentally requires sufficient Sulf1 activity to maintain a tight control over VEGF activity. VEGF induction in Sulf1 overexpressing HMec1 cells was also accompanied by increased expression of multiple co-receptors and ligands, such as NRP1 and angiopoietin, although VEGFR1 and VEGFR2 activities were barely detectable under these conditions. Sulf1 over-expression inducing increased VEGF expression also led to increased HS6ST1 activity as inferred from increased mRNA and protein expression. These data imply that the combined activities of Sulf1 and sulfotransferase activities regulate the net VEGF activity in HMec1 cells. It is unclear, however, why Sulf2 over-expression with a similar reported 6-O sulfate de-sulfation activity did not also induce VEGF-A in HMec1 cells. This clearly highlights distinct roles for Sulf1 and Sulf2 in regulating VEGF cell signalling in dermal endothelial cells. Sulf1 and Sulf2 transfected HMec1 cells also exhibited markedly different cell morphologies as Sulf1 transfection led to endothelial cell aggregation, supporting the value of further investigation into their distinct roles in endothelial cell differentiation and cell aggregation under normal and pathological conditions.

Unlike in HMec1 cells, Sulf1 over-expression in ea926 cells, in contrast, showed the opposite effect on VEGF induction; control ea926 cells already express higher levels of VEGF at baseline, compared with HMec1, and these were reduced following Sulf1 over-expression, but largely unaltered upon Sulf2 over-expression. Taken together, these differences emphasise the highly context-dependent pro- and anti-angiogenic Sulf activities in different endothelial cell types. Sulf1 and Sulf2 differences observed in both HMec1 and ea926 cell transfections may relate to their regulatory roles, rather than simple cell signalling facilitation or inhibition by heparan sulfate 6-O sulfate hydrolysis. Unlike HMec1 cells, Sulf1/Sulf2 effects on ea926 cells are more compatible with the reported anti-angiogenic activity of Sulf1 [6,19,28], but potentially suggests a pro-angiogenic activity for Sulf2. It, however, remains to be determined whether HMec1 and ea926 responses to Sulf1 and Sulf2 over-expression vary due to their different endothelial cell origins from microvasculature versus macrovasculature characteristics, due to arteriovenous differences or due to immortalisation altering the phenotype of ea926 cells. This analysis is thus limited by a restriction of Sulf1/Sulf2 responses in only two fairly diverse endothelial cell lines that need to be extended to primary endothelial cells and more physiological cardiovascular cell lines.

Sulf1 overexpression in HMec1 cells not only selectively induced VEGF expression but also induced selective changes in the levels of TGFβ cell signalling components that included Smad2, Smad3 and ALK5, despite these cell signalling components being barely detectable in control or Sulf2 transfected HMec1 cells. Unlike VEGF-A, the role of TGFβ1, a multi-functional cytokine in angiogenesis, however, is less well defined in normal cells and is believed to act as an indirect angiogenic agent, at least in some cancer cells [29]. Some of the ambiguity of its role in angiogenesis may also relate to its ability to bind to the TGFβ type II receptors via two different type I receptors, ALK1 and ALK5, with opposing activities in endothelial cells. ALK1 is believed to promote angiogenesis by stimulating BMP cell signalling through the activities of Smad1/5 transcription factors, whereas ALK5 is known to inhibit angiogenesis by TGFβ cell signalling through the activities of Smad2/3 transcription factors inhibiting endothelial cell proliferation [30]. Sulf1 over-expression in HMec1 cells selectively induced TGFβ signalling and barely detectable levels of BMP signalling and may thus explain some inhibition of endothelial cell proliferation and cell migration, despite VEGF expression. A reduction in HMec1 endothelial cell proliferation and migration, however, was also apparent with Sulf2 over-expression, despite the barely detectable levels of Smad2/3 and ALK5, suggesting that in this case, lack of VEGF activity may be a limiting factor. Thus, there is not a simple, straightforward relationship to angiogenesis through relative activities of TGFβ and BMP cell signalling alone without impact on VEGF cell signalling. TGFβ and VEGF signalling showed a very close relationship to each other in HMec1 endothelial cells, mutually induced or non-responsive upon Sulf1 or Sulf2 over-expression, respectively, with largely similar responses upon stimulation by TGFβ1 or VEGF-A. VEGF-A and TGFβ1 thus appear to regulate each other’s activities by changing their expression patterns during both Sulf1 and VEGF over-expression associated with hypoxia. This implies that SULFs have a much greater regulatory role than merely cell signalling facilitation by regulating HSPG 6-O sulfation extracellularly; rather there is also a subsequent feedback mechanism regulating HS 6-O sulfotransferase activities.

Not only did Sulf1 and Sulf2 demonstrate opposite effects on both TGFβ and VEGF signalling, but they also demonstrated cell line-specific effects on these cell signalling pathways, with Sulf1 over-expression promoting both TGFβ and VEGF signalling in HMec1 cells, and attenuating signalling in ea926 cells. Sulf1 over-expression in ea926 cells also clearly inhibited BMP signalling, as observed from the marked down-regulation of Smad1/5 and ALK1 activities. The relationship between BMP signalling and cell proliferation and migration was less clear as both proliferation and migration were reduced not only in Sulf1 over-expressing cells but also in Sulf2 over-expressing ea926 cells, despite the latter maintaining both BMP and VEGF signalling.

It remains to be determined how the upstream activity of TGFβ regulates VEGF expression and how the responses of these pathways are closely interlinked. This was apparent not only by induction of TGFβ and VEGF upon Sulf1 over-expression in HMec1 cells but also following exposure to exogenous TGFβ or VEGF. The lack of increased VEGF induction in Sulf2 over-expressing HMec1 cells may explain reduced cell proliferation and cell migration due to increased cell differentiation highlighted by reduced cell size of Sulf2-over-expressing HMec1 cells and increased expression of some junctional proteins (unpublished). This, however, differs from in vivo speculation of the Sulf2 enzyme having some pro-angiogenic activity in some cancers [24], although cancer angiogenic cells may also have some additional mutations modifying their normal cell signalling responses.

RT PCR analysis, despite being semiquantitative, demonstrated significant differences in multiple cell signalling pathway components more clearly than immunocytochemical analyses. Protein level differences resulting from Sulf over-expression were more readily detectable by immunocytochemistry in HMec1 cells than in ea926 cells since mRNA changes in these cells were modest.

Sulf1 and Sulf2 enzymes are believed to edit HSPG 6-O sulfation to counter the action of HS 6-O sulfotransferases, but the present study shows that Sulfatases also selectively regulate HS6 sulfotransferase activities themselves, presumably via a feedback mechanism, rather than relying entirely on extracellular editing. In addition, this study also demonstrated marked Sulf1 and Sulf2 functional differences in their regulation of VEGF and TGFβ cell signalling activities in an endothelial cell type-specific manner. In view of their identical HSPG 6-O de-sulfation capacity, it is difficult to imagine how Sulf1 and Sulf2 enzymes may facilitate marked differences in VEGF and TGFβ cell signalling to achieve selective feedback regulation of HS 6-O sulfotransferase activities without exploiting some additional unique properties of these enzymes that may be explained by closer examination of Sulf1 versus Sulf2 functional microheterogeneity. This study also highlights the much greater complexity of developing target-specific therapeutics, as it is not just VEGF but multiple inputs that regulate the net cell signalling activity impacting angiogenesis in specific endothelial cell populations. Characterisation of Sulf1/Sulf2 functional heterogeneity has clinical implications not only in angiogenic cell signalling but also in the prevention or reduction of thrombosis and atherosclerosis, which requires further studies.

## 4. Materials and Methods

### 4.1. Cell Culture and Sulf1/Sulf2 Transfection

Two human endothelial cell lines, HMec1 (ATCC) and Ea926 (ATCC), were used to examine the effect of Sulf1 and Sulf2 overexpression. Ea926 cells were grown in Dulbecco’s modified Eagle’s (DMEM) medium with 10% foetal calf serum (FCS), while HMec1 cells were grown in MCDB131 medium (ThermoFisher Scientific, UK) supplemented with glutamine, EGF (10 ng/mL), hydrocortisone (1 μg/mL) and 10% FCS. Over-expression of Sulf1 or Sulf2 variants in these cells was achieved using BioRad transfectin reagent (Bio-Rad Laboratories Ltd, Maxted Road, Hemel Hempstead, HP2 7DX, UK) as previously described [21] that also included transfection with an EGFP expression vector or pcDNA3 for use as controls. To eliminate any untransfected cells, G418 (Sigma Aldrich, UK) was added to the growth medium following 48 h of growth in the normal medium [21]. Such transfected cells were used for all in vitro analyses following 2–6 weeks of growth in G418 containing medium.

For in vitro proliferation assays, 10,000 cells/well were seeded in multiple 24-well plates and fixed for SRB analyses following 1 day, 3 day and/or 4 days growth. Cultured cells on plates were fixed using trichloroacetic acid and stained with Sulforhodamine B, washed and dried before the bound dye was solubilized in 10 mM unbuffered Tris base and the absorbance measured at OD 565 nm [26]. The amount of dye extracted provides a measure of cell mass and the number of cells in a sample if the cells are approximately similar in size. In addition to each sample consisting of a minimum of three technical replicates for each stage, each experiment was repeated three times.

For cell migration assays, 5000 cells were seeded into each insert of an Ibidi Culture dish with 3-well silicone inserts for overnight growth. The insert was removed once the cells were confluent the next day and imaged at 0 and 8 h or also at 20 h time intervals as appropriate for different cell lines.

### 4.2. RT PCR Analysis

Total RNA was prepared from control (EGFP and pcDNA3 transfected cells) and Sulf1/Sulf2 transfected cells using Trizol (Invitrogen) according to the manufacturer’s instructions. The RNA integrity was assessed by electrophoresis using Syber safe staining and OD260/OD280 nm absorption ratio (>1.95) [31]. Total RNA (1 μg) for each sample was reverse-transcribed into cDNA with SuperScript II reverse transcriptase (Invitrogen) using random primers (Invitrogen) for RT-PCR analysis with 35 PCR amplification cycles using the following primers: HS6ST1: 5′-CATCACCCTGCTACGAGACC-3′ & 5′-AAGGGCCGGATGAACTTGAG-3′; HS6ST2: 5′- CCAAGTCAAATCTGAAGCACA-3′ & 5′-CTGGAAATGGGTCTGAAGGA-3′; HS6ST3: 5′- CTTGCGGGAGTTTATGGATTG-3′ & 5′- GGTGCTCTAGCTGCTTGGTGT-3′; Smad2: 5′-AACAGAACTTCCGCCTCTGG-3′ & 5′-ACCGTCTGCCTTCGGTATTC-3′; Smad3: 5′-GGAGACACATCGGAAGAGGC-3′ & 5′-CCCTCCCCATCCCAAGTCTA-3′; ALK5: 5′-GGGGCGACGGCGTTACAGTGTTTCTGCCAC-3′ & 5′-TGAGATGCAGACGAAGCACACTGGTCCAGC-3′; TGFβ1:5′-AAGTGGATCCACGAGCCCAA-3′ & 5′-GCTGCACTTGCAGGAGCGCAC-3′; TGFβRII: 5′-CACCGCACGTTCAGAAGTC-3′ & 5′-GAGGCTGATGCCTGTCACTT-3′; Smad1: 5′-GAGACAGCTTTATTTCACCATATCC-3′ & 5′-CATAGTAGACAATAGAGCACCAGTGTTTT-3′; Smad5: 5′-CGGTAGCCACTGACTTTGAGTTAC-3′ & 5′-AGCTGAAATGGACTTCCTGGTC-3′; ALK1:5′-CCTTGCTGGCCCTGGTGGCCCT-3 & 5′-GTGGGCAATGGCTGGTTTG-3′; VEGF: 5′- GAAGTGGTGAAGTTCATGGATGTC-3′ & 5′-CGATCGTTCTGTATCAGTCTTTCC-3′; VEGF-A: 5′-CATGAACTTTCTGCTGTCTTGG-3′ & 5′- CCTGGTGAGAGATCTGGTTCC-3′; VEGFR1: 5′-CTGAGAACAACGTGGTGAAGATT-3′ & 5′-CTGACATCATCAGAGCTTCCTGA-3′; Ang1: 5′-GGAAGTCTAGATTTCCAAAGAGGC-3′ & 5′-CTTTATCCCATTCAGTTTTCCATG-3′; Ang2: 5′- GCCACAACCATGATGATCC-3′ & 5′-TTCTTGGTTGTGACAGCAGC-3′; NRP1:5′-GGCTCCAAATAGACCTGGGG-3′ & 5′-GGTGCTGTCTATGACCGTGG-3′; NRP2: 5′-GGATGGCATTCCACATGTTG-3′ & 5′- ACCAGGTAGTAACGCGCAGAG-3′; Tie1: 5′-CAAGGTCACACACACGGTGAA-3′ & 5′-GCCAGTCTAGGGTATTGAAGTAGGA-3′; Tie2: 5′-TGCCCAGATATTGGTGTCCT-3′ & 5′-CTCATAAAGCGTGGTATTCACGTA-3′; Sulf1: 5′-CCGATGATCAAGATGTGGAGC-3′ & 5′-GCATTGGTCCTGTGTACT GC-3′; Sulf2: 5′-CAGGTTTCAGAGGGACCGCAG-3′ & 5′-GAGTCGTCCACCGACATGAGG-3′; β-actin: 5′-CTATGAGCTGCCTGACGGTC-3′ & 5′-AGTTTCATGGATGCCACAGG-3′. RT PCR fragments were separated in 1% agarose gels using β-actin fragment to normalise the sample loading. All RT PCR experiments were repeated at least three times. Expression levels were quantified using Image J and statistical analysis was preformed using ANOVA, where data depicting a *p* value < 0.05 were considered statistically significant.

### 4.3. Immunocytochemistry

HMec1 and ea926 cells grown on Lab-Tek glass slides were fixed in 4% paraformaldehyde for 15 min before staining with different antibodies as previously described [27]. This involved incubation of fixed cells with permeabilization buffer for 15 min at room temperature followed by incubation with 10% foetal calf serum (FCS) for 30 min before treatment with different primary antibodies at different dilutions, e.g., SULF1 (1/200), SULF2 (1/100) and as dilutions recommended by Suppliers for all antibodies. The binding of all the rabbit primary antibodies was detected using streptavidin Alexa Fluor 594 or Alexa Fluor 488 fluorochrome bound to biotin-linked goat anti-rabbit immunoglobulins as previously described [32]. The binding of mouse monoclonal antibodies was detected using goat anti-mouse immunoglobulins linked to Alexa Fluor 488 or Alexa Fluor 594 fluorochrome. Sections treated with pre-immune rabbit and mouse sera were similarly incubated with fluorochrome-labelled secondary antibodies as controls (not shown). All primary antibody incubations overnight at 4 °C were followed by secondary antibody incubations for 1 h each at room temperature. Following four PBS washes between and after each incubation, labelled tissue sections were mounted in a polyvinyl alcohol mounting medium with DABCO and 2.5 μg/mL DAPI for nuclear visualisation to photograph images using a Leica DM4000B fluorescent microscope.

## Figures and Tables

**Figure 1 ijms-23-03769-f001:**
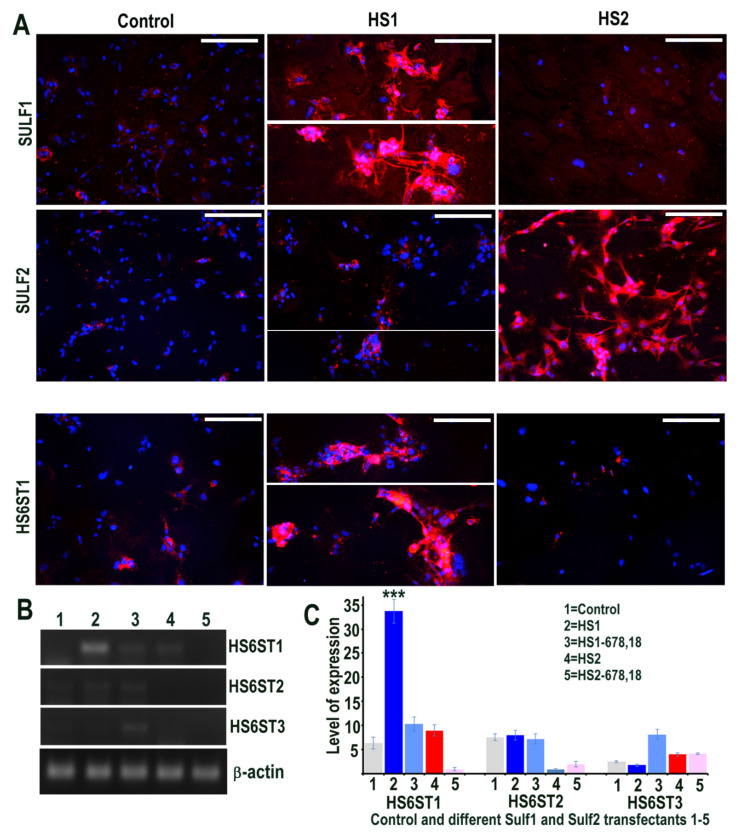
(**A**) Levels of Sulf1 and Sulf2 protein expression in HMec1 cells transfected with pcDNA3 (control) and Sulf1 (HS1) and Sulf2 (HS2) genes examined by antibodies to SULF1 and SULF2 using immunofluorescence procedure including the changes in the levels of HS6 sulfotransferase 1 following such transfections. Scale bar = 100 μM. (**B**,**C**) Selective changes in the levels of HS6ST1, HS6ST2 and HS6ST3 were also examined using RT PCR analysis and quantified using Image J analysis. Error bars = means ± SD; *** *p* < 0.0001.

**Figure 2 ijms-23-03769-f002:**
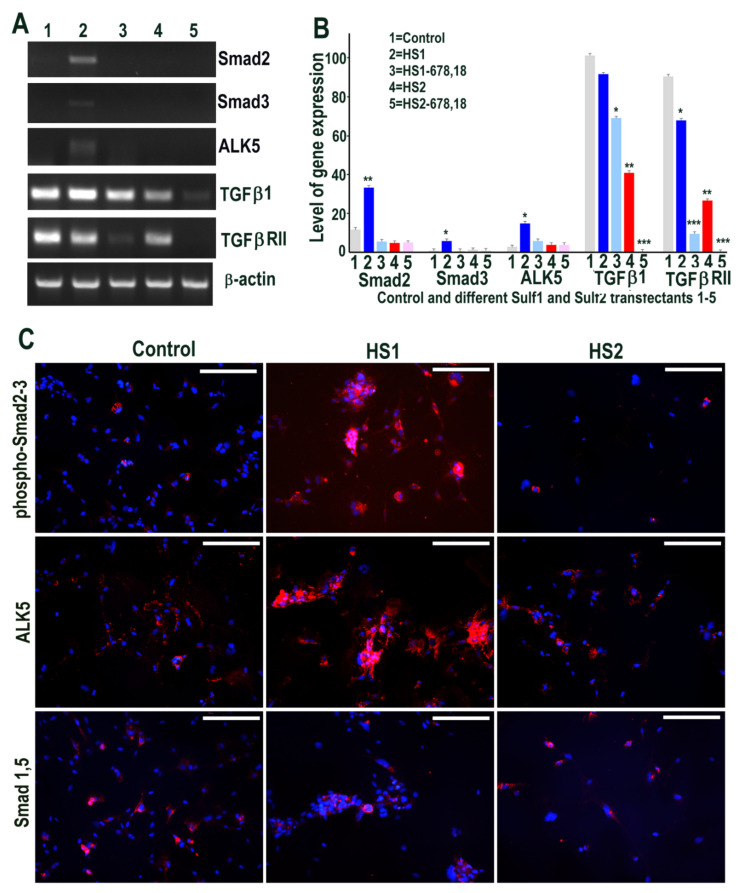
(**A**,**B**) Selective changes in levels of TGFβ cell signalling following transfections of HMec1 cells with (1) pcDNA3, (2) full length Sulf1 (HS1), (3) shorter inactive Sulf1 (HS1-678,18), (4) full length Sulf2 (HS2) and shorter inactive Sulf2 (HS2-678,18) examined by RT PCR analysis and quantified using Image J analysis. Error bars = means ± SD; *** *p* < 0.0001; ** *p* < 0.001; * *p* < 0.01. (**C**) Changes in TGFβ cell signalling following transfections were also confirmed by immunostaining for phosphoSmad2/3 and ALK5 as well as very low level detection with no apparent changes in BMP signalling using antibody to Smad1/5 that also confirmed virtually no expression using RT PCR procedure (not shown). Scale bar = 100 μM.

**Figure 3 ijms-23-03769-f003:**
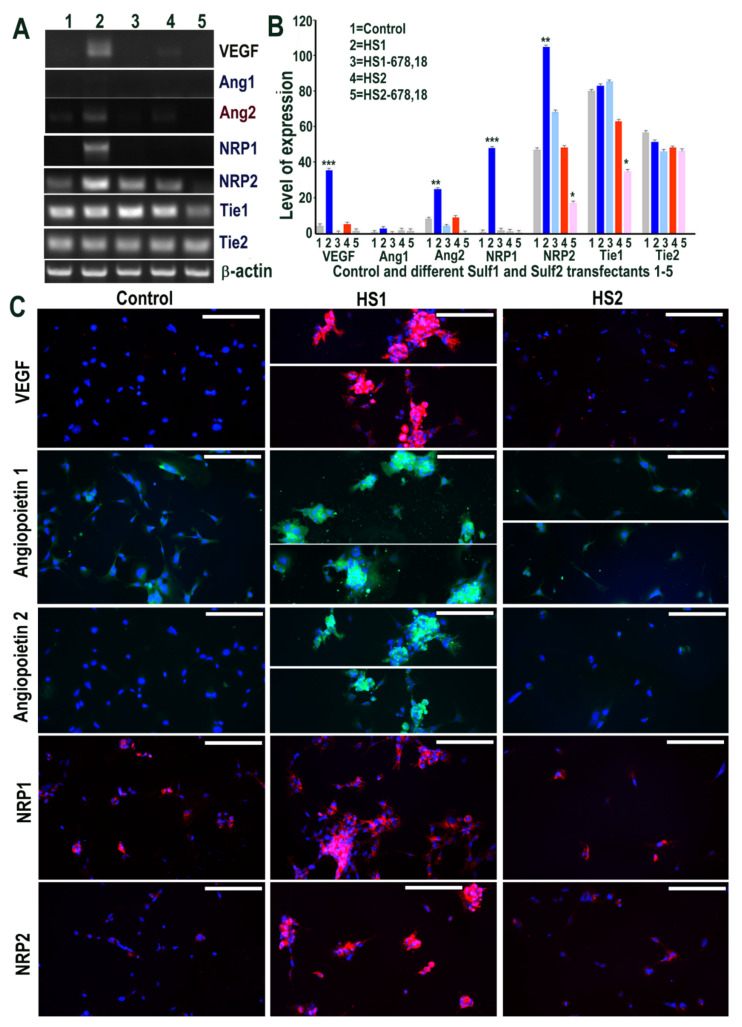
(**A**,**B**) Selective changes in levels of VEGF cell signalling including its receptors (Ang1, Ang2, NRP1, NRP2, Tie1 and Tie2) following transfections of HMec1 cells with (1) pcDNA3, (2) full length Sulf1 (HS1), (3) shorter inactive Sulf1 (HS1-678,18), (4) full length Sulf2 (HS2) and shorter inactive Sulf2 (HS2-678,18) examined by RT PCR analysis and quantified using Image J analysis. Error bars = means ± SD; *** *p* < 0.0001; ** *p* < 0.001; * *p* < 0.01. (**C**) Changes in VEGF cell signalling following transfections were also confirmed by immunostaining for VEGF, Ang1, Ang2, NRP1, NRP2 using immunofluorescence procedure. Scale bar = 100 μM.

**Figure 4 ijms-23-03769-f004:**
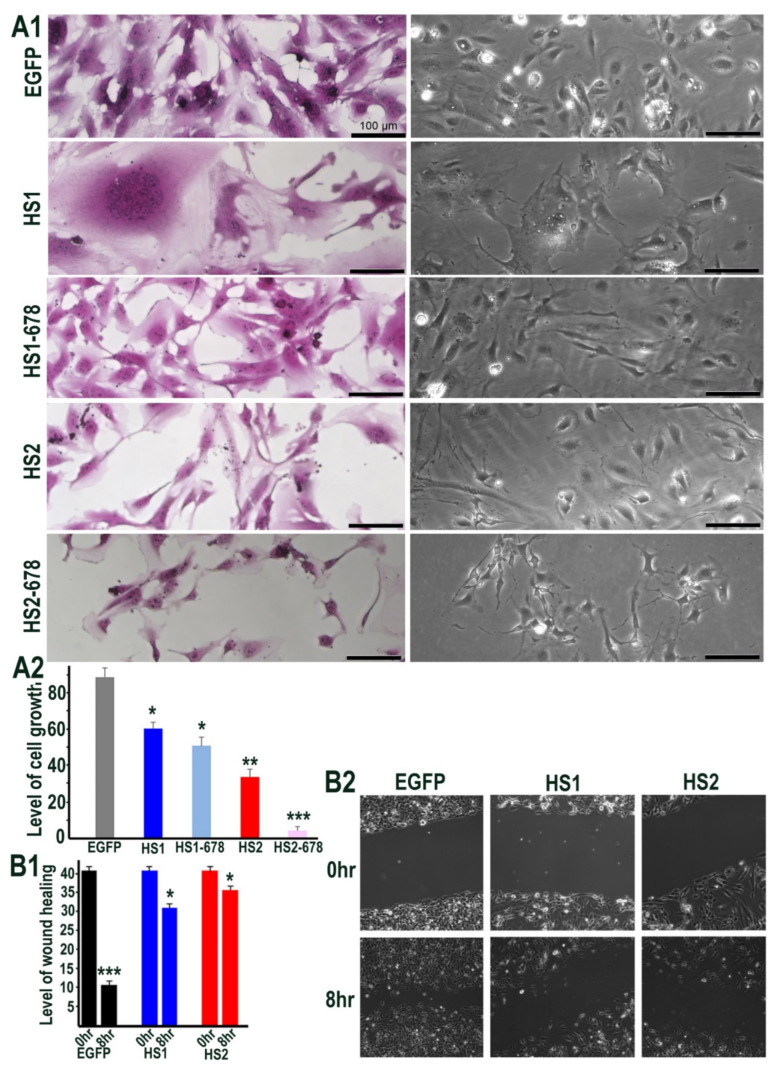
(**A1**) Cell morphology of HMec1 cells transfected with EGFP, HS1 (full length Sulf1), HS1-678 (shorter inactive Sulf1), HS2 (full length Sulf2), HS2-678 (shorter inactive Sulf2) examined by phase contrast microscopy of cells (right lane) and photographing cells following staining with SRB dye (left lane). Scale bar: 100 μM. (**A2**) represents the changes in cell proliferation or relative cell mass of transfected and control cells following absorbance measurements of SRB stain at OD 565 nm. (**B1**,**B2**) represent the changes in wound healing/cell migration in control (EGFP) versus Sulf1 (HS1) and Sulf2 (HS2) transfected cells. Error bars = means ± SD; *** *p* < 0.0001; ** *p* < 0.001; * *p* < 0.01.

**Figure 5 ijms-23-03769-f005:**
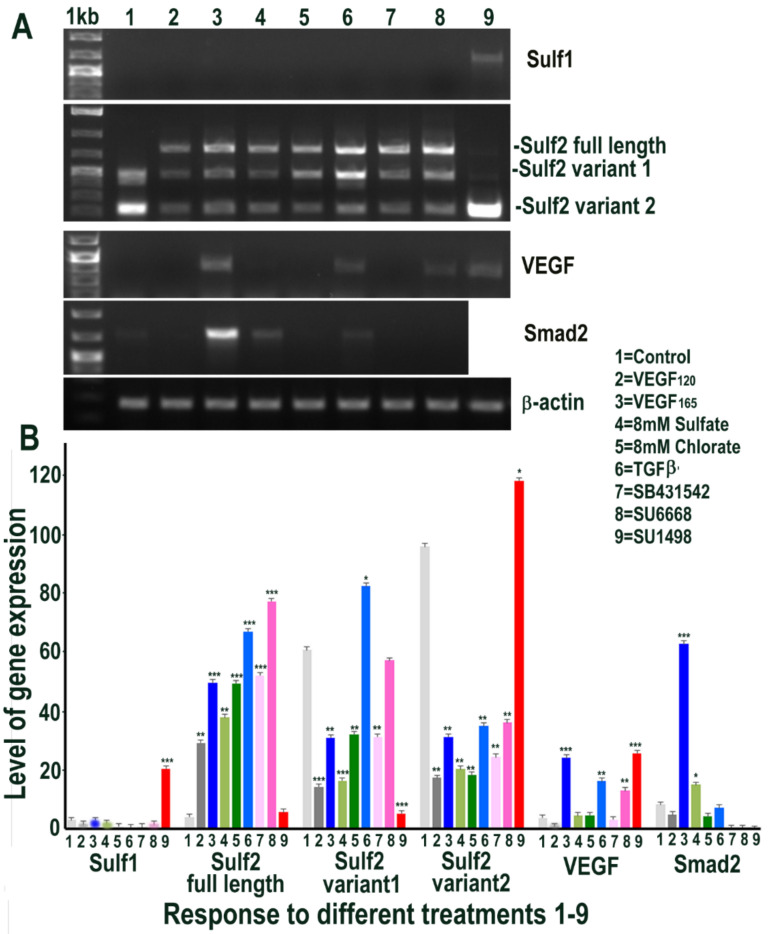
RT PCR analysis of HMec1 cells (**A**) and its quantification of Sulf1, Sulf2, VEGF and Smad2 (**B**) in untreated control (1) and following treatments with VEGF_120_ 50 ng/mL (2) VEGF_165_ 50 ng/mL (3) 8 mM sodium sulfate (4) 8 mM sodium chlorate (5) TGFβ1 10 ng/mL (6) 10 µM SB431542, TGFβ inhibitor (7) 5 µM SU6668, antiangiogenic RTK inhibitor (8) and 10 µM SU1498, VEGFR2 inhibitor (9) for 48 h. Error bars = means ± SD; *** *p* < 0.0001; ** *p* < 0.001; * *p* < 0.01.

**Figure 6 ijms-23-03769-f006:**
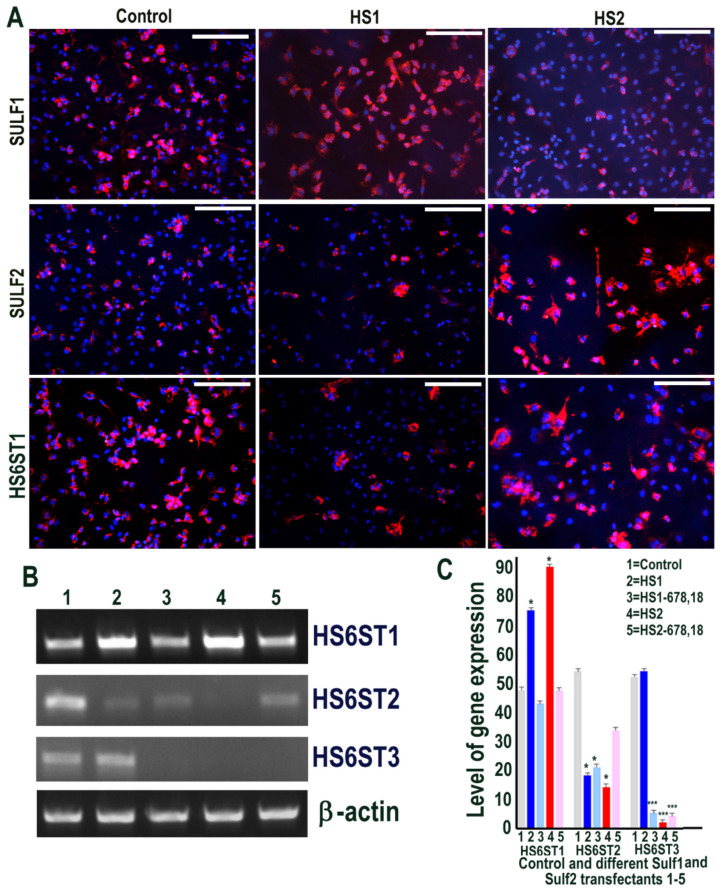
(**A**) Levels of SULF1 and SULF2 proteins in ea926 cells transfected with pcDNA3 (control) and Sulf1 (HS1) and Sulf2 (HS2) genes examined by antibodies to SULF1 and SULF2 proteins using immunofluorescence procedure including the changes in the levels of HS6 sufotransferase 1 following such transfections. Scale bar: 100 μM. (**B**,**C**) Some changes in the levels of HS6ST1, HS6ST2 and HS6ST3 were also apparent using RT PCR analysis quantified using image J analysis. Error bars = means ± SD; *** *p* < 0.0001; * *p* < 0.01.

**Figure 7 ijms-23-03769-f007:**
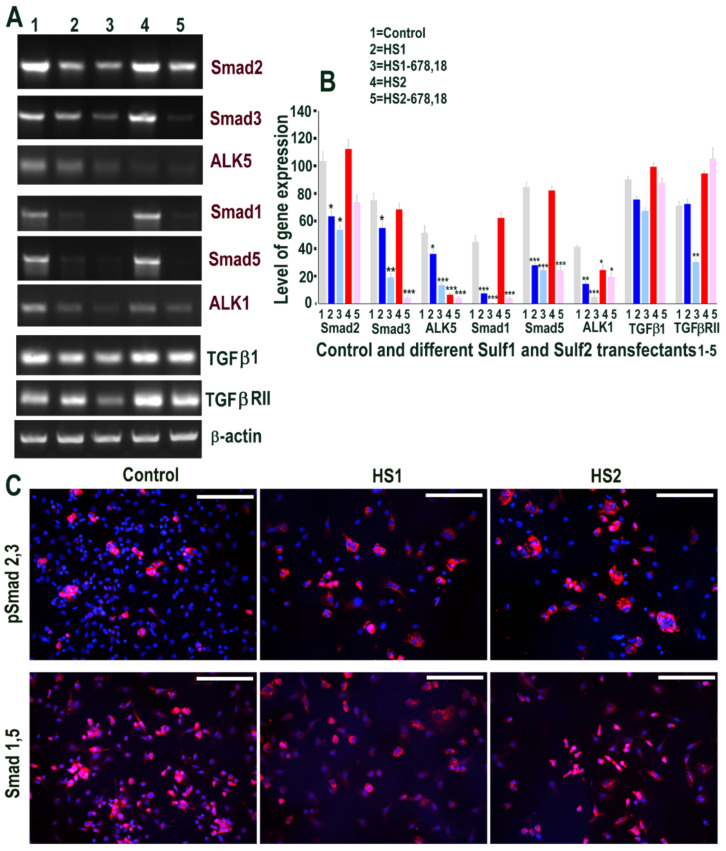
(**A**,**B**) Some changes in levels of TGFβ & BMP cell signalling following transfections of HMec1 cells with (1) pcDNA3, (2) full length Sulf1 (HS1) and (3) shorter inactive Sulf1 (HS1-678,18), (4) full length Sulf2 (HS2) and shorter inactive Sulf2 (HS2-678,18) examined by RT PCR analysis quantified using image J analysis. Error bars = means ± SD; *** *p* < 0.0001; ** *p* < 0.001; * *p* < 0.01. (**C**) Changes in TGFβ & BMP cell signalling following control, Sulf1 and Sulf2 transfections confirmed by immunostaining with phosphor Smad2,3 and Smad1,5 antibodies. Scale bar: 100 μM.

**Figure 8 ijms-23-03769-f008:**
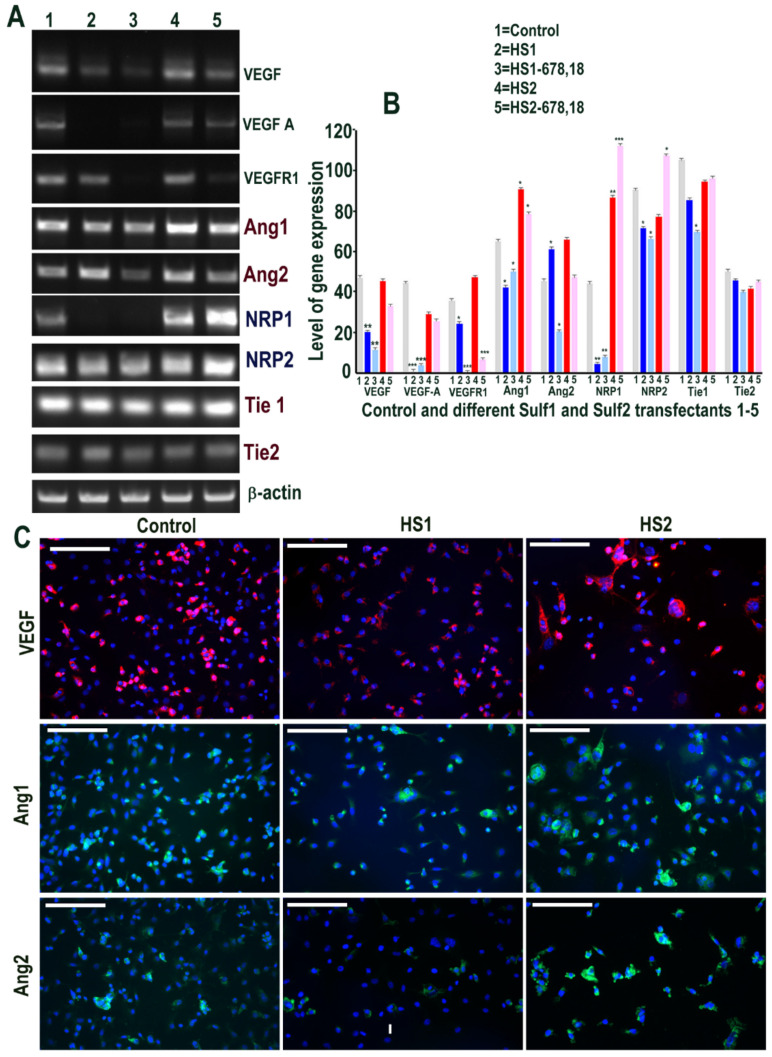
(**A**,**B**) Changes in levels of VEGF cell signalling including its receptors (Ang1, Ang2, NRP1, NRP2, Tie1 and Tie2) following transfections of ea926 cells with (1) pcDNA3, (2) full length Sulf1 (HS1), (3) shorter inactive Sulf1 (HS1-678,18), (4) full length Sulf2 (HS2) and shorter inactive Sulf2 (HS2-678,18) examined by RT PCR analysis quantified using image J analysis. Error bars = means ± SD; *** *p* < 0.0001; ** *p* < 0.001; * *p* < 0.01. (**C**) Changes in VEGF cell signalling following transfections were also examined by immunostaining for VEGF, Ang1 and Ang2 using immunofluorescence procedure. Scale bar: 100 μM.

**Figure 9 ijms-23-03769-f009:**
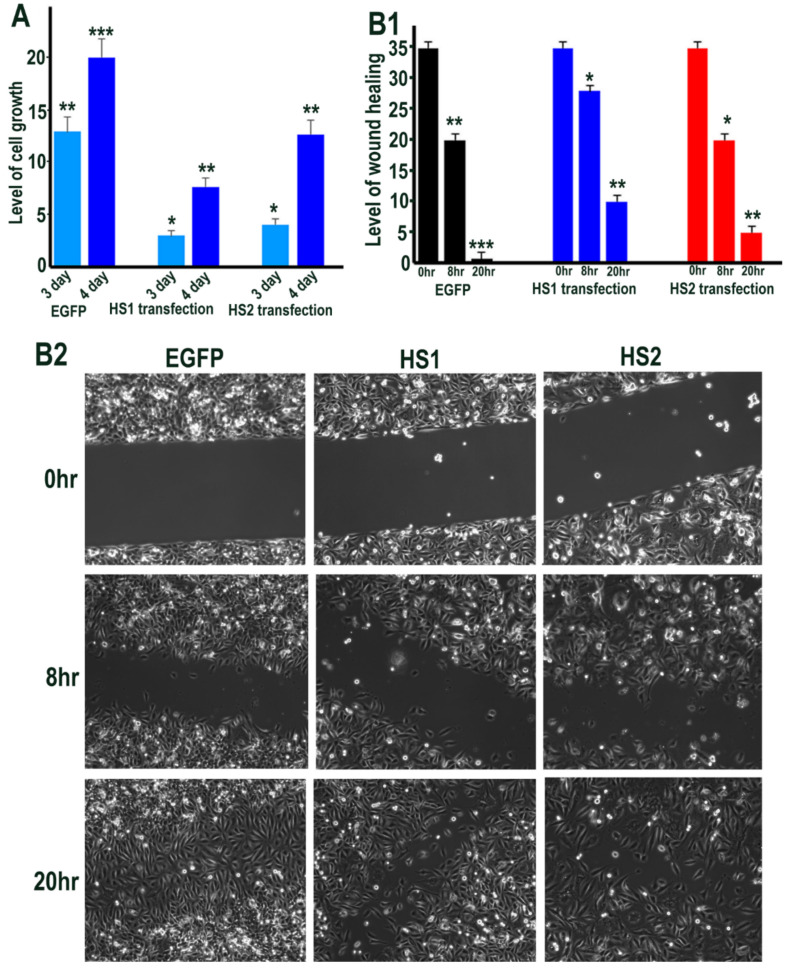
(**A**) Represents the changes in cell proliferation or relative cell mass of EGFP (control), HS1 (Sulf1) and HS2 (Sulf2) transfected ea926 cells following absorbance measurements of SRB stain at OD 565 nm after 3 or 4 days of growth by comparing growth at day 1. (**B1**,**B2**) represent the changes in wound healing/cell migration in control (EGFP) versus Sulf1 (HS1) and Sulf2 (HS2) transfected cells at 8 and 20 h following insert removal. Error bars = means ± SD; *** *p* < 0.0001; ** *p* < 0.001; * *p* < 0.01.

**Figure 10 ijms-23-03769-f010:**
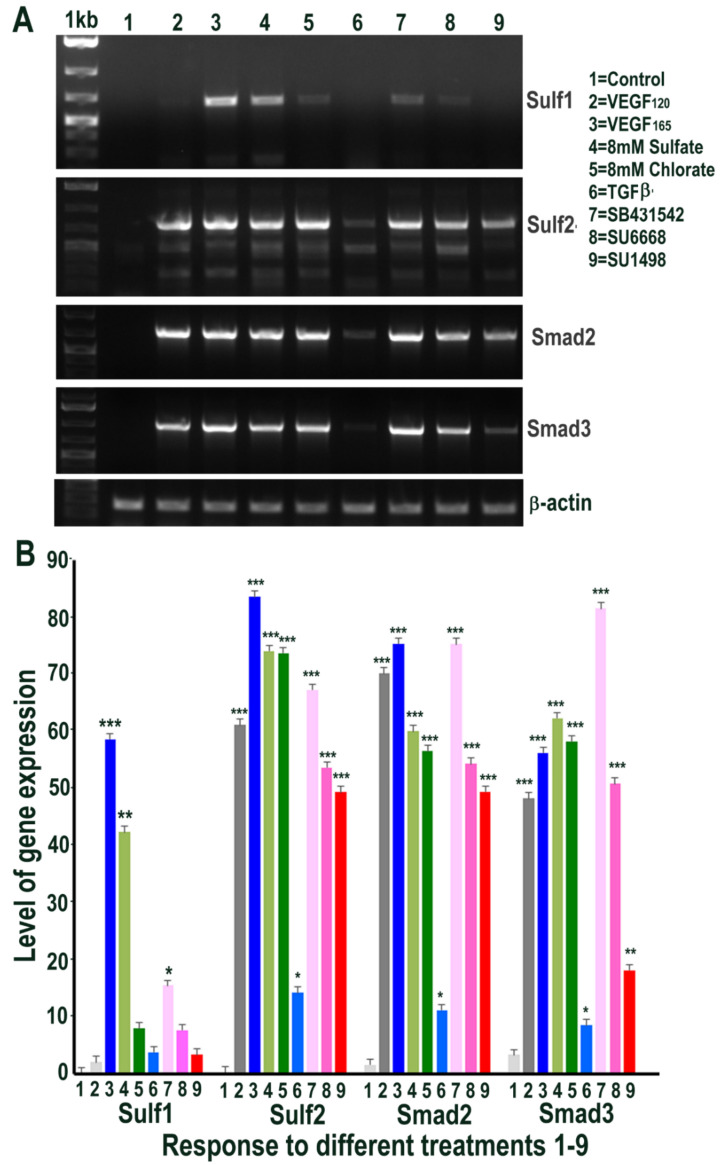
RT PCR analysis of ea926 cells (**A**) and its quantification of Sulf1, Sulf2, Smad2 and Smad3 (**B**) in untreated cells (1) and following treatments with 50 ng/mL VEGF_120_ (2) 50 ng/mL VEGF_165_ (3) 8 mM sodium sulfate (4) 8 mM sodium chlorate (5) 50 ng/mL TGFβ (6) 10 µM SB431542, TGFβ inhibitor (7) 5 µM SU6668, antiangiogenic RTK inhibitor (8) and 10 µM SU1498, VEGFR2 inhibitor (9) for 48 h. Error bars = means ± SD; *** *p* < 0.0001; ** *p* < 0.001; * *p* < 0.01.

## Data Availability

Not applicable.

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
