# Peer review of "Context Dependent Sulf1/Sulf2 Functional Divergence in Endothelial Cell Activity"

_ijms, 2022, doi:10.3390/ijms23073769_

Round 1

Reviewer 1 Report

This is an interesting experimental study focused on a largely unexplored issue. Nevertheless, I strongly suggest that the authors might mention at least some potential clinical implications of their findings.

Reviewer 2 Report

Dhoot et al reported that roles for Sulf1 and Sulf2 in modulating VEGF cell signaling in human endothelial cells. This study was well designed, data are sound. Results provide new information in the pathophysiology of cardiovascular diseases. There are several issues that need to be revised or addressed. A revision is suggested.

  1. Please briefly address the methods and conclusion in the abstract.
  2. Please strengthen the role of endothelial cell activity in human cardiovascular diseases. Also, please emphasize this study's motivation and present the knowledge gap according to previous studies.
  3. Please provide scale bars for every photo.
  4. RNA expression should also be confirmed by protein expression.
  5. How to transfer the findings to clinical implications? Please discuss.
  6. Please discuss the limitation of this study.
  7. Different photo sizes were present ( such as Fig 4B2). Please be consistent.
  8. How to make sure the same location or place was taken in wound healing assay? Please provide another assay to confirm the conclusion.
  9. Dosage of treatment condition should be present in Fig. Leg.

Author Response

Response to referee report 2

Round 2

Reviewer 2 Report

My questions had been addressed, this submission is acceteable.